# Nodular Cast Iron GGG40, 60, 70 Mechanical Characterization from Bars and Blocks Obtained from Brazilian Foundry

Daniel de Oliveira Fernandes [1,*], Carla Tatiana Mota Anflor [1], Jhon Nero Vaz Goulart [1] and Besim Baranoğlu [2]

1    Group of Experimental and Computational Mechanics, University of Brasilia, Gama, Brasilia 72405-610, DF, Brazil; anflor@unb.br (C.T.M.A.); jvaz@unb.br (J.N.V.G.)
2    Metal Forming Center of Excellence, Atilim University, Kızılcaşar Mahallesi 06836, Turkey; besim.baranoglu@atilim.edu.tr
*    Correspondence: danieldeoliveirafernandes@gmail.com

**Abstract:** Nodular cast iron has been commonly applied in industry and many engineering applications due to its low production cost and the similarity of its mechanical properties to carbon steel. The mechanical properties of nodular cast iron are very dependent on its microstructure and also on the characteristics of the graphite nodules. In this sense, the main objective of this paper was to evaluate and characterize the nodular cast iron grades GGG40, GGG60 and GGG70 in the absence of heat treatment. In addition, specimens were obtained from casted bars and blocks without the Y-block casting process. The microstructure was analyzed by optical microscopy with the support of computational image analysis for determination of the attributes of the graphite nodules and the quantification of each phase present in the microstructure of the nodular cast iron. The results showed that the microstructure has a strong effect on the material's strength, especially the density of graphite nodules in the material. This difference reinforces the idea that cast iron can undergo mechanical changes due to changes in the casting process, confirming the importance of checking the characteristics of the cast batch before engineering applications of the material.

**Keywords:** nodular cast iron; mechanical characterization; microstructure; computational image analysis



## 1. Introduction

Nodular cast iron is an alternative to commercial carbon steels due to the similarity of their mechanical properties and the low production cost in relation to steel. NCI has good machinability and is approximately 10% lighter than the steel [1]. According to [2], the present NCI production cost varies at around 20% to 40% less than commercial steel. This type of iron has a lower production cost because of the synthetic melting process, which replaces some of the expensive pig iron with cheaper scrap iron. Normally, silicon and carbon are added with the aim of obtaining a higher nodule number.

NCI is not a single material but a class of materials offering a wide range of properties obtained through microstructure control [3]. NCI presents the graphite in a crack-arresting nodule shape, making it ductile. The spherical shape tends to have a lower stress concentration ($\sigma_{max}/\sigma_{med}$ = 1.7), while lamellar and flake graphite results in higher stress ($\sigma_{max}/\sigma_{med}$ = 5.4) [4].

The mechanical properties of NCI are strongly dependent on its microstructure [5–7]. The metal matrix can be composed mainly of ferrite, resulting in low strength values associated with high ductility and toughness values. In a ferritic matrix, NCI presents a strength limit of 350–450 MPa, associated with 10–22% elongation. Another composition is that the matrix is constituted by pearlite, which implies good mechanical strength values associated with relatively low ductility values. In a pearlitic matrix, the NCI strength limit can reach 900 MPa associated with 2% elongation, and by then producing mixtures of ferrite and pearlite, different classes of cast iron are obtained, with diverse combinations of properties, each suitable for a specific application [8].

Due to a combination of excellent properties, the nodular cast iron family has been increasingly applied in various engineering fields and has become a research material [9]. Given the properties and machinability of this material, NCI has been replacing grey cast iron, malleable cast iron, cast and forged steel, and welded structures [10]. Typical applications of nodular cast iron mainly include components such as pulleys, shafts, sprockets, valves, and hydraulic components, pinions, gears, bearings, brake calipers, and supports, crankshafts, camshafts, and suspension parts of vehicles, among others.

The NCI manufacturing process comprises many variables, each of which affects the final material's mechanical properties. Graphite has low mechanical strength when compared to the metallic matrix [11]. The presence of graphite can be regarded as a mechanical discontinuity and a stress concentration point in the matrix. In addition, the graphite shape also has a marked influence on the material's mechanical properties. All the aforementioned characteristics intensify the search for understanding the mechanical properties of NCI and their variations. The properties of NCI are strictly dependent on its material microstructure, size, shape, and nodule distribution, and the presence of defects resulting from the manufacturing process has a direct impact on the material properties. The authors in [12] investigated the mechanical properties depending on the cross-section thickness in GGG40 NCI, and the authors in ref. [13] evaluated the effects of microstructure, mechanical and physical properties on machinability of graphite cast irons. Samec et al. [14] analyzed the low cycle fatigue behaviour of NCI GGG50 subject to high temperatures of 300 °C and 400 °C, with applications in railway brake disks.

It is worth mentioning that several studies concern NCI obtained from Y-blocks or U-blocks, a different scenario from the small and medium foundries. Important engineering components are manufactured from casted materials, and with the increased use of NCI, special attention must be paid to the mechanical and microstructural properties of components cast directly in blocks, bars, or in the final shape component. As is well-known, NCI has a range of mechanical properties that vary greatly with the cast control, leakage temperature, cooling rates, chemical composition, and cast shape. The mechanical properties of nodular cast irons are intimately related to their microstructure, and parameters such as the nodule count, nodularity, and phase content are preponderant factors in the final mechanical properties [15,16]. Despite the mechanisms regarding the microstructure being well-known in the literature, the academic community is aware that the mechanical properties may differ widely from the usual parameters specified in the standards. These deviations in mechanical and microstructure properties ensure that NCI becomes an open research subject.

A good support that has been used in microstructural analysis is image segmentation. Image processing techniques have been widely employed (applied) in many fields, such as robot automatisation [17], vehicle detection [18], medical procedures and diagnosis [19,20], and in the and in the analysis of mechanical materials [21] and NCI chacterization [22].

Image segmentation allows for obtaining the measurement, count, and size of the graphite nodules. Segmentation based on thresholding of greyscale images makes it possible to binarise the image. In many cases, this is enough for evidencing the domains [23]. Computational image analysis can be a robust tool with the capacity to define the cast iron class in the presence of dust, scratches, and measurement noise [24].

The present work aims to present a comprehensive study concerning the mechanical behavior of commercial NCI GGG40, GGG60, and GGG70, targeting the Brazilian industrial problem of large variations in mechanical properties of the obtained cast iron. As aforementioned, the nodular cast irons present a wide variety of mechanical properties due to several factors resulting from the casting process. In this sense, the efforts rely on the material characterization of blocks and bars from the same batch. The study was carried out through mechanical tests and microstructural analysis using micrograph images with the aid of digital image segmentation.

## 2. Materials and Methods

This section will present the procedures used in the development of this work. Mechanical tests such as a tensile test hardness test were performed based on the test standard. In order to complement the material characterization, mechanical tests were performed, together with microscopy analysis, allowing for the (proper) identification of the nodules and phases of the NCI. With the microstructure analysis, a MATLAB routine capable of counting and characterising the nodules was developed.

### 2.1. Materials

For the development of this research on the mechanical and metallographic analysis of nodular cast iron, 3 different classes of nodular cast irons were required: GGG40, GGG60, and GGG70 cast in round bar and block shape. The material was purchased from a medium-sized Brazilian foundry.

The division was carried out so that the influence of the initial geometry on the mechanical properties of the manufactured specimen could be verified. The aim was to analyze the microstructure and its influence on the material strength according to the material ordered as cast. The chemical composition was measured at a certified laboratory using the JEOL JSM 6610 SEM. The procedure followed the ASTM E1508 [25] and E766 [26] recommendations. Table 1 presents the chemical composition of the GGG40, 60, and 70 used in this research.

**Table 1.** Chemical composition of GGG40, 60 and 70.

| Chemical Composition (%) | C | Mn | Si | S | P | Mg | Cu | Cr |
|---|---|---|---|---|---|---|---|---|
| GGG40 | 3.54 | 0.20 | 2.30 | 0.011 | 0.060 | 0.038 | 0.090 | - |
| GGG60 | 3.68 | 0.30 | 2.39 | 0.022 | 0.070 | 0.045 | 0.055 | 0.080 |
| GGG70 | 3.45 | 0.31 | 2.88 | 0.030 | 0.075 | 0.045 | 0.75 | 0.010 |

The material analysis depends on the cast block and bar shape. The bar has a 1 meter length per 2 inches of diameter, and the cast iron blocks have dimensions of $300 \times 150 \times 150$ (in mm). The specimens were arranged in the block in such a way it is possible to evaluate the amount of graphite and its influence in each section. This methodology is similar to that adopted by [27]. The specimens were properly identified and then subjected to mechanical tests. The division of the material block and the arrangement of the specimens in it are shown in Figure 1.

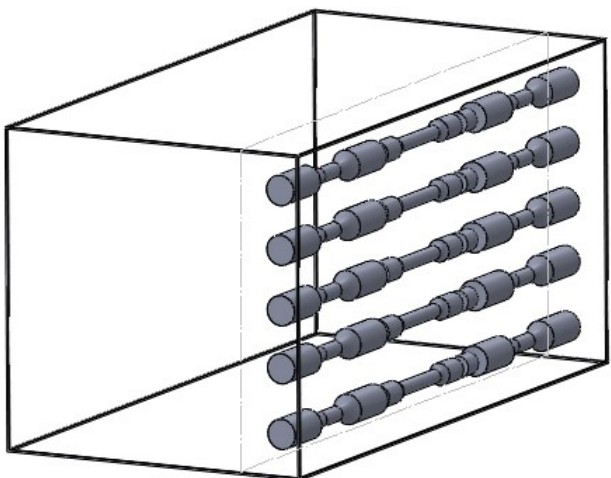

**Figure 1.** Isometric view of the arrangement of the specimens in the solid material block.

### 2.2. Tensile Test

The tensile test is based on the ASTM E8 [28], and from this test we can obtain important quantitative and qualitative data on the resistance, stiffness and ductility of the material tested. Tensile tests for all specimens were performed at a displacement speed of 1 mm/min. The stop criterion chosen was the differential ratio loading, set to 40%, which is a safe value to work at; when reached, this means that the specimens have ruptured.

### 2.3. Hardness Test

The Vickers hardness test performed was based on the (ASTM E92, 2017) [29] recommendations. A ZHU250 ZwickRoell was used for the universal hardness test with a force of 50 kgf (HV50). For the universal hardness test, five measurements were performed in each specimen in the case of the round bar sample, represented by Figure 2, and another five measurements were performed at each point of the sample block, represented by Figure 1.

The microhardness test was performed to obtain the ferrite, pearlite, and graphite hardness. The test consists of the use of a calibrated machine to force a Diamond indenter into the surface of the material being evaluated. In this test, the DuraScan G5—EMCO-TEST for microindentation hardness was used. The test forces range from 1 to 1000 gf, and the diagonals are measured after load removal. A 0.05 kgf force was applied in a microindentation test and five measurements were performed in each phase of the nodular cast iron for each sample. For any microindentation hardness test, it is assumed that the indentation does not undergo elastic recovery after force removal. The test followed the (ASTM E384, 2017) [30] recommendations.

The specimens were the same as those used for the metallographic procedure. After the image acquisition for metallographic analysis, the specimens were subjected to the hardness test. First, the microhardness procedure was performed, and subsequently, the universal hardness test.

### 2.4. Bauschinger Effect Test

The Bauschinger effect tests were carried out with the Instron 8801 and tests were strain-controlled. To reveal the cyclic characteristics of each material, the specimens tested were obtained from the casted block and round bar. Specimens were manufactured according to [31]. Undesirable buckling could appear during testing in the compression loading phase. The authors in Ref. [31] mention the ratio between the radius-of-curvature and the minimum radius-of-specimen. Lower ratio limits will increase the stress concentration and may affect the fatigue life; higher ratios limit the specimen's buckling resistance.

### 2.5. Metallographic Procedure

The metallographic analysis procedure in the nodular cast irons GGG40, GGG60 and GGG70 was carried out following the (ASTM E3, 2012) [32] recommendations. The metallographic analysis of the circular bars consists of removing the bar ends, thus forming two 2-inch-diameter samples. For the GGG40 case, these samples were named with suffix-1 representing a top sample and suffix-2 representing a bottom sample, e.g., 40-1 (top) and 40-2 (bottom). The purpose of this methodology is to quantify and characterize the graphite nodules in 9 different positions in the specimen and evaluate the nodule characteristic in relation to the sample position in the round bar, the top being the nearest to the pouring section. The marking of where the optical micrograph should be performed was divided as follows: in the center of the sample, 4 divisions within a radius of 10 mm and 4 divisions within a radius of 20 mm. Five measurements were performed in each demarked zone. These samples were not etched, as the objective here is to detect the nodule characteristics in the round bar position. Figure 2 shows a round bar end metallographic sample demonstration.

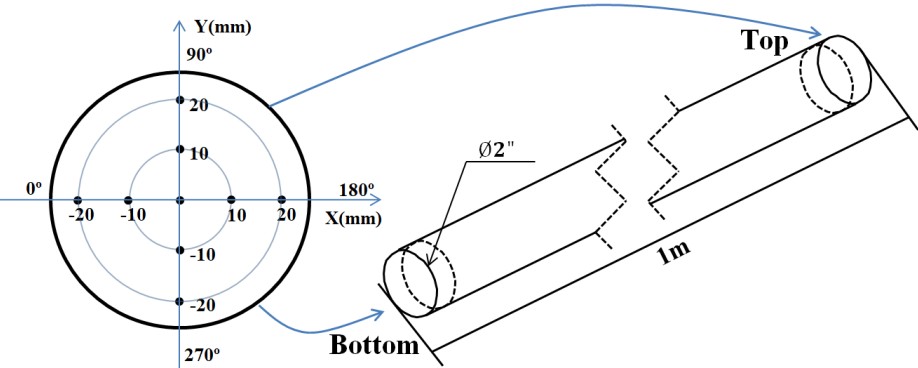

**Figure 2.** Round bar end metalographic sample marking demonstration.

Metallographic specimens were also removed from the tested tensile specimens obtained from the block and the bar. The specimens were cut and mounted in a resin and then ground and polished. For each metallographic sample, five micrograph images were performed. To ensure proper graphite retention, attention must be paid to the loading applied over the specimen; the proper control of these factors influences the graphite retention. Examination of the material microstructure was performed with a trinocular inverted metallographic microscope. The specimens were etched with a 2% nital solution (alcohol + $HNO_3$), by immersion for up to 5 s. Microstructural analysis was carried out in order to obtain the percentage of ferrite, pearlite and graphite.

### 2.6. Computational Image Analysis

A routine was developed using the considerations given by (ASTM E2567, 2015) [33] and (ASTM A247, 2020) [34], which is the standard for determining the nodularity and nodule count in ductile cast iron using image analysis. Figure 3 shows an input software image.

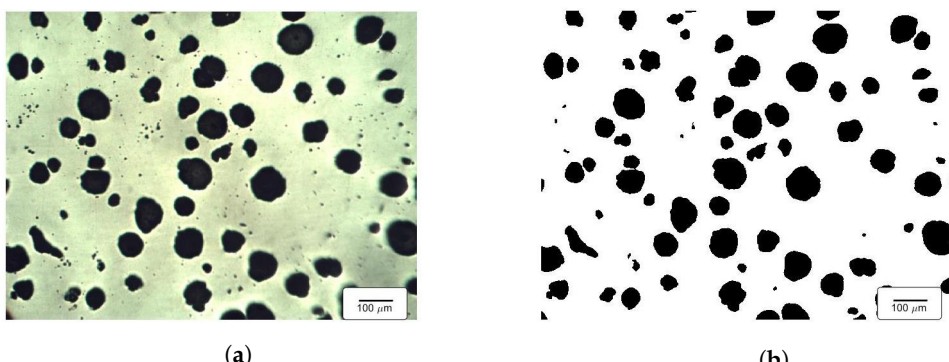

(**a**)          (**b**)

**Figure 3.** Sample microstructure image of nodular cast iron. 100× magnification. (**a**) Input image. (**b**) Image segmentation.

The graphite nodules have the characteristic of being darker than the metal. In this stage, the chemical attack was not carried out in order to identify the graphite nodules easily. If there no adjustments in this step, the software can binarize the areas that do not contain nodules and count as nodules. This is due to the threshold value, for which, as a rule, 50 was used, which means that every pixel that has information above 50 is considered white and every pixel that has information below 50 is considered black. As a result of the light adjustment when the micrograph image was achieved, some images may be darker than others. Adjustment of the threshold value must be performed manually, so the software considers as black only the area that is graphite. Figure 4 outlines the total procedure for the nodule count.

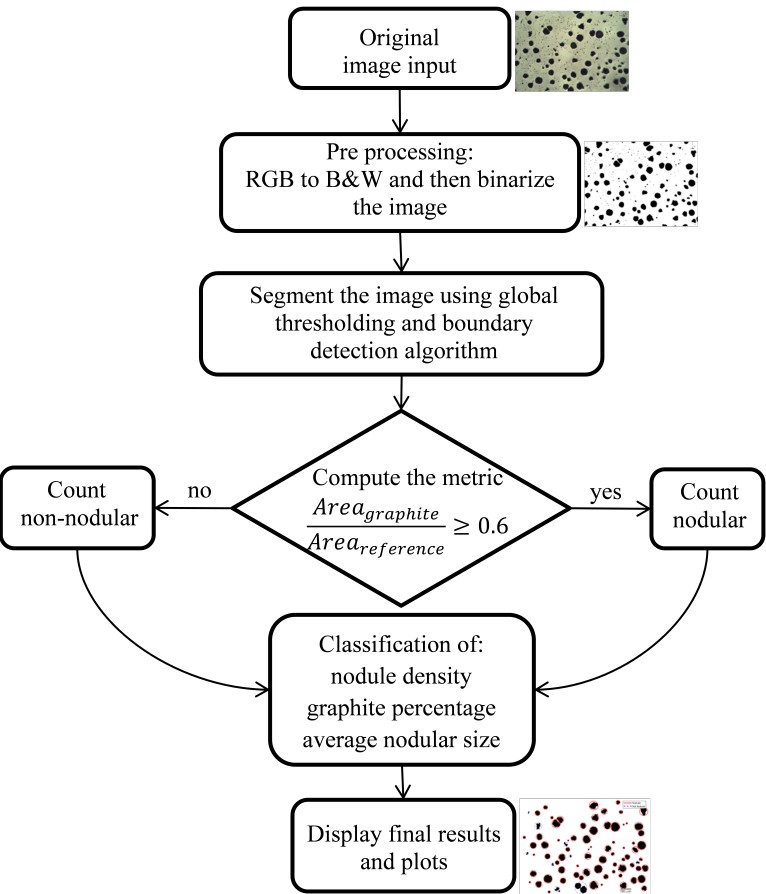

**Figure 4.** Nodule count algorithm flowchart.

After the binarization process, the image contains a white background with the sets of black pixels that form an object, in this case, a graphite particle. The standard recommends that particles smaller than 10 μ should be excluded as a way to remove noise. In addition, particles that touch the edges of the micrograph image are excluded and only particles that are complete in the image are considered for the evaluation process.

When the image is segmented, each nodule can be well-identified and useful morphological properties can be easily identified. The set of black pixels is identified through the boundary-detection algorithm.

The calculations were developed according to ASTM E2567 [33]. To define if a particle can be considered circular or not, parameters such as roundness shape factor, compactness, sphericity and eccentricity are used. The parameter used in this work was the roundness shape factor. An example of RSF is shown in Figure 5.

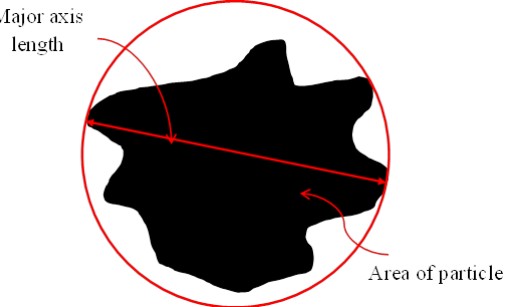

**Figure 5.** Schematic representation of the roundness shape factor.

The minimum required shape factor value to qualify a particle as being nodular is suggested by [33] to be 0.60, and this was the threshold value adopted.

The developed routine was used to evaluate the graphite nodule density, graphite average area, percentage of nodular graphite, sphericity, compactness, eccentricity, graphite, ferrite and pearlite content percentage in etched samples. The developed routines are available on the web for nodule counting and characterization [35] and phase counting [36].

## 3. Results and Discussion

First, the results regarding the mechanical tests: tensile test and hardness test. From these results, an analysis of the nodule characteristics and quantity is performed.

In the second part, the casted round bars are analyzed by computational image analysis, where the nodule density and graphite percentage on certain areas of the round bar cross-section are verified.

Finally, the third part presents the microstructure with chemical etching and computational image analysis, complementing the discussion of the results.

### 3.1. Mechanical Tests

Figure 6 show the stress–strain curve and a comparison between the GGG40, 60 and 70 tested specimens.

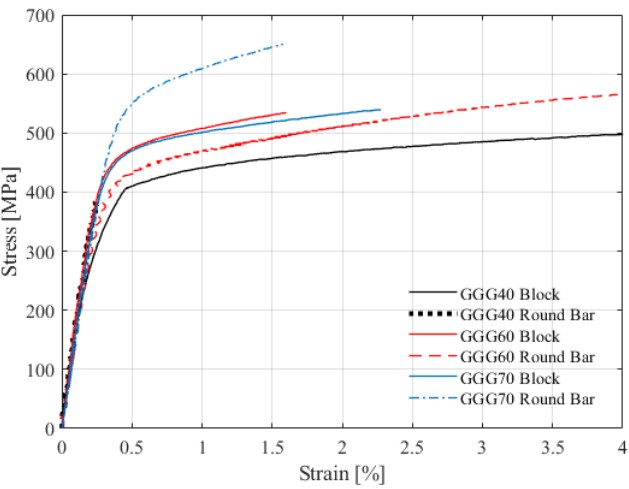

**Figure 6.** Tensile test comparison.

GGG40 round bar samples did not reached the yield limit, displaying a premature rupture. This behavior was repeated along the round bar tensile test. Small air inclusions in the samples made from the bar were present, which caused a weakening in the material tested. In general, specimens from the block have greater elongation than the round bar specimens. For NCI GGG60, this phenomenon has not occurred due to the higher concentration of ferrite phase in round bar samples. The GGG70 round bar sample has a higher yield limit than the block specimen, and this could be explained by the pearlite content being grater in round bar samples. The elastic modulus fits in with the standard values between 154 and 180 GPa. The mechanical properties obtained from the specimens removed from the block differ from the standard data obtained using the Y-block procedure. This difference in the obtained data could have been due to the fact that specimen extraction was performed without using the Y-block.

From a Bauschinger effect test, we obtained the test data for performing curve fitting through the Ramberg–Osgood equation. Figure 7 shows the comparison between the monotonic and cyclic behavior.

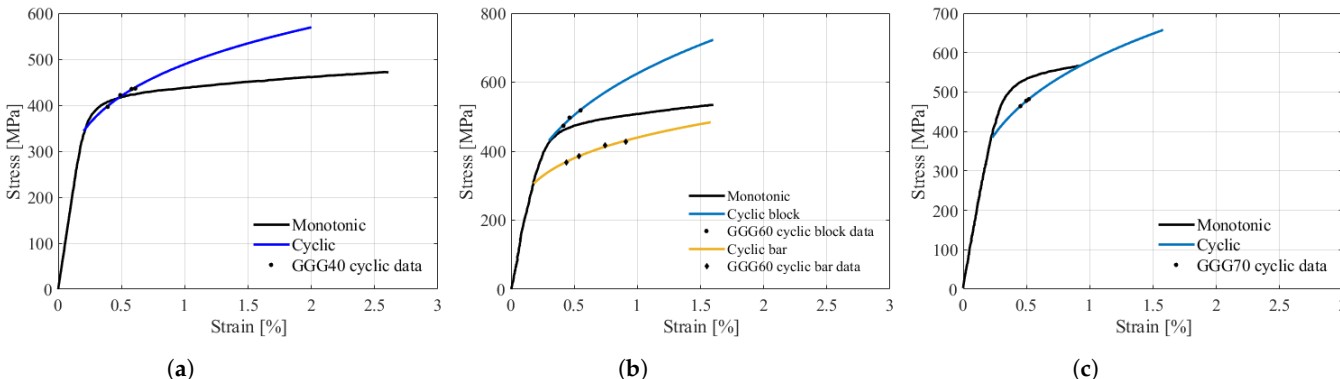

**Figure 7.** Cyclic and monotonic stress-strain curve for (**a**) GGG40 (**b**) GGG60 (**c**) GGG70.

It can be noted that GGG40, GGG60, and GGG70 under cyclic conditions tend to have a hardening behavior. The behavior concerning the specimens from the round bar was the same as that in the block specimens, except for the GGG60 round bar specimens. The high ferrite amount in the GGG60 round bar samples leads the material to display a softening behavior. The Ramberg–Osgood coefficients found in the curve adjustment are available in Table 2.

**Table 2.** Ramberg–Osgood curve-fitting results for GGG40, 60 and 70.

| | Ramberg–Osgood Coef. | | |
|---|---|---|---|
| | *K* | *n* | *R²* |
| GGG40 | 489.0 | 0.2190 | 0.9617 |
| GGG60 block | 625.1 | 0.3090 | 0.9617 |
| GGG60 bar | 439.4 | 0.2106 | 0.9776 |
| GGG70 | 578.9 | 0.2782 | 0.9921 |

Table 3 presents the microindentation test results for GGG40, 60 and 70.

**Table 3.** Microindentation test results.

| | | Hardness (HV) | | |
|---|---|---|---|---|
| | | **Ferrite** | **Pearlite** | **Graphite** |
| GGG40 | Block | 212 ± 2.1% | 306 ± 7.6% | 114 ± 4.0% |
| | Bar | 222 ± 3.5% | 383 ± 2.8% | 94 ± 7.5% |
| GGG60 | Block | 230 ± 6.7% | 339 ± 5.3% | 69 ± 4.1% |
| | Bar | 226 ± 3.0% | 386 ± 2.2% | 87 ± 8.1% |
| GGG70 | Block | 246 ± 4.3% | 372 ± 5.9% | 65 ± 4.4% |
| | Bar | 224 ± 3.1% | 444 ± 6.2% | 100 ± 7.6% |

From the results presented in Table 3, one can note that the samples removed from the bar have pearlite hardness superior to that of the samples removed from the block. This is due to the geometry section being thinner and the cooling rate being lower. The cooling rate increase promotes pearlite hardness. As the cooling rate increases, the carbon diffusion rate decreases, and consequently the interlamellar distance decreases. It is of note that the hardening behavior in graphite particles must be the same for the pearlite phase, in that as the carbon diffusion decreases, graphite becomes harder. The general hardness of the GGG60 round bar however was not superior to block samples. These values could be explained by the material microstructure. Round bar samples have a large amount of ferrite, and graphite particles have a large number with vermicular and non-nodular shapes, which was not expected in this type of material.

Figure 8 shows the influence of the graphite nodule density at maximum stress in GGG40, 60 and 70.

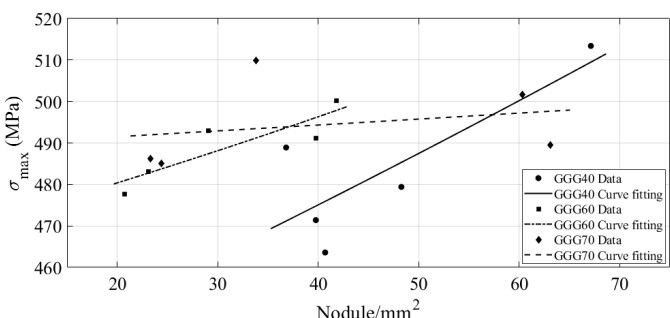

**Figure 8.** Relation between nodule density and maximum stress in GGG40, 60 and 70.

The plot demonstrates that the higher the nodule density, the higher will be the strength. This consideration means that smaller graphite nodules could contribute to the NCI mechanical strength. In accordance with GGG40, in GGG60, this effect was seen to be more aggressive, which could be related to the average nodule size in the sample. For NCI GGG70, the obtained data have some dispersion due to specimens 2 and 3 failing before reaching the yield limit. Even with the dispersion of data, the tendency to have greater resistance with a greater density of nodules was repeated.

### 3.2. Round Bar Ends Analysis

Figure 9 shows the nodule count per mm$^2$ at specific points in the cast iron round bar cross-section. Two samples were analyzed, one from the top and the other from the bottom, as explained in Section 2.5. Figure 10 shows the percentage of nodular graphite in relation to all graphite particles. Both figures represent NCI GGG40 results.

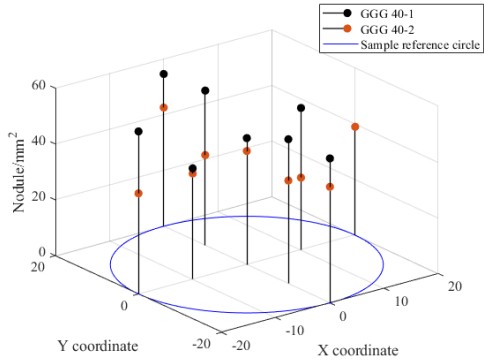

**Figure 9.** Nodule count in GGG40.

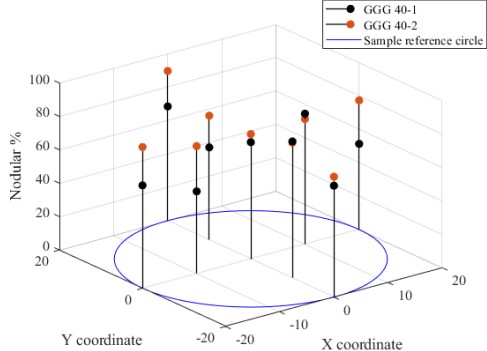

**Figure 10.** Graphite percentage in GGG40.

Figures 11 and 12 present the results for the nodule count and graphite percentage for NCI GGG60.

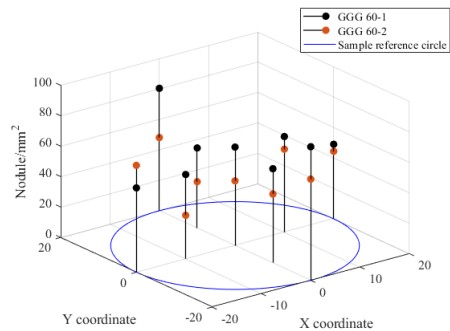

**Figure 11.** Nodule count in GGG60.

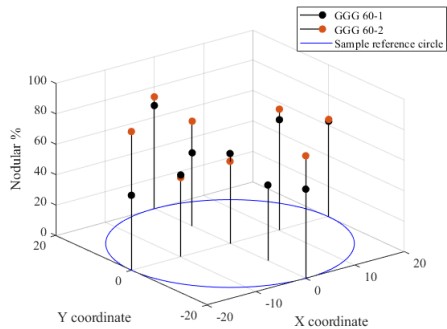

**Figure 12.** Graphite percentage in GGG60.

Figures 13 and 14 present the results for the nodule count and graphite percentage for NCI GGG70.

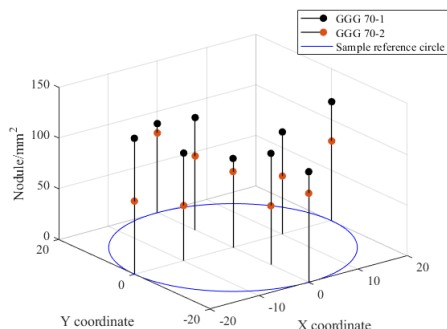

**Figure 13.** Nodule count in GGG70.

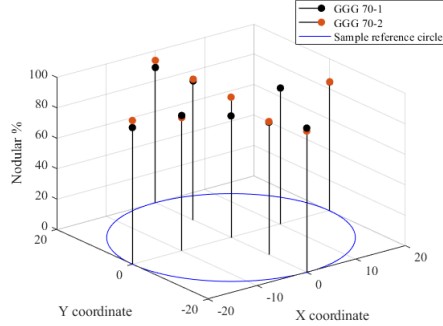

**Figure 14.** Graphite percentage in GGG70.

It is possible to observe a pattern of behaviour in the presented charts. The samples with index indicator −1 correspond to the specimen removed from the top of the round casted bar. This means that the graphite concentration is higher at the top of the casted material. As the graphite nodule is less dense than the material, the nodules tend to float to the surface, causing a higher nodule concentration per area unit in the sample. However, the percentage of graphite particles with a spherical shape was higher in the bottom samples.

The difference was slightly higher in the percentage of nodular graphite, and this phenomenon can be explained by the cooling rate being slightly higher than at the top (leakage of material); so, at the bottom the cooling rate is lower, which helps in the nucleation of the graphite.

Another consideration is that the density of nodules recorded in the measurement at the centre of the sample is lower than that measured near the edge of the sample. The explanation is that in the geometric edges, the cooling rate is higher than in the middle, so the time for graphite nucleation is not enough. As a logical explanation, the average graphite size tends to be greater in the sample centre because of the low cooling rates, which enable graphite nucleation. The greater the graphite nodule size, following a proportional logic, the lower will be the nodule density measured. The curious fact is that this phenomenon does not occur.

The average graphite nodule size was lower in the centre of the sample in comparison with the edges of the samples. It is possible to see that the nodular graphite percentage in the middle was lower than at the edges. The graphite nodules do not meet the circularity factors, which affects the count and the final size of the nodules of the analysis carried out in the centre of the sample. This effect can be explained by the shrinkage allowance, which is the contraction of the final volume after solidification. The analyzed samples were taken from the tops of the bar where this phenomenon is more pronounced, affecting the formation of nodules in the centre of the sample.

Figures 15–17 show the hardness test results from the bar samples. The measurements were taken from the center, four points in the inner radius, and four points in the outer radius, as demonstrated in Figure 2. Five measurements were performed at each point in order to obtain the average of the measurements and the final hardness result value at the given point.

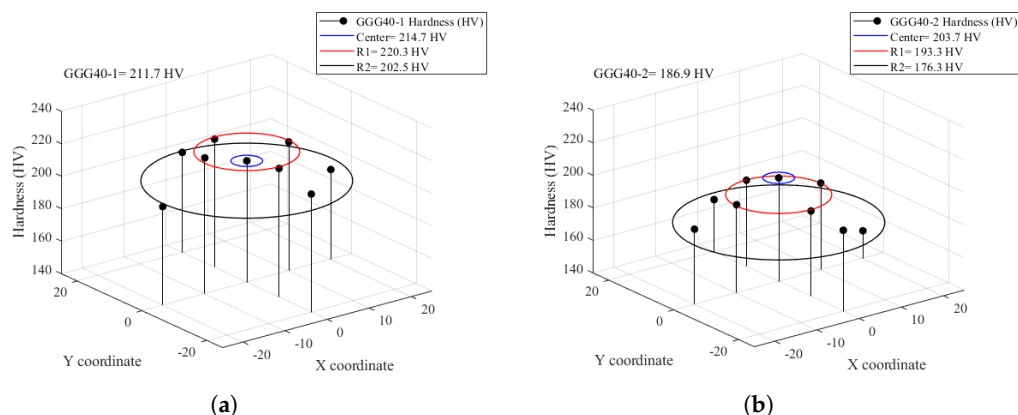

(**a**)          (**b**)

**Figure 15.** Hardness test round bar sample. (**a**) GGG40-1. (**b**) GGG40-2.

The results show that the samples with −1 index have higher hardness compared with the −2 index samples. This could be explained by the fact that −2 index samples have higher graphite nodules in the samples. The graphite particles are much softer than the ferrite or pearlite matrix, so bigger nodules could affect the material decreasing the hardness.

From Figures 18–20, it is possible to see the analyzed samples after image segmentation. The routine considered a nodular, graphite particle that fits the parameters established in Section 2.5.

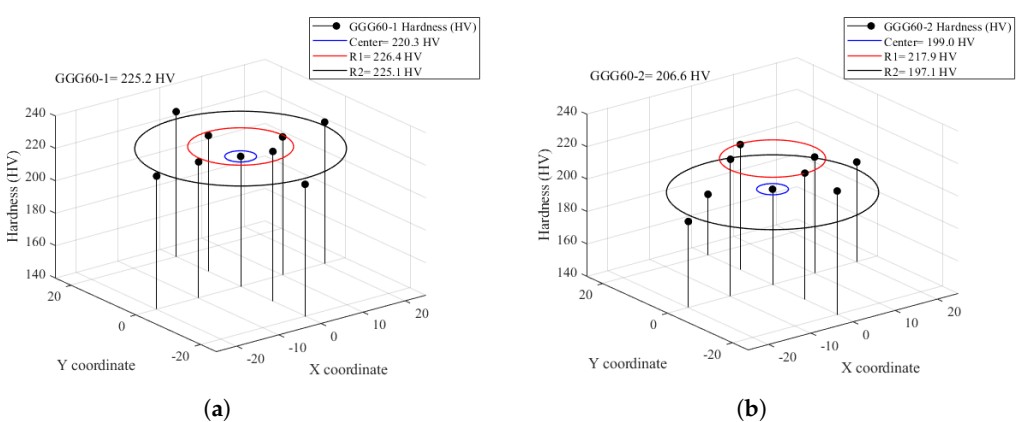

(**a**)                                        (**b**)

**Figure 16.** Hardness test round bar sample. (**a**) GGG60-1. (**b**) GGG60-2.

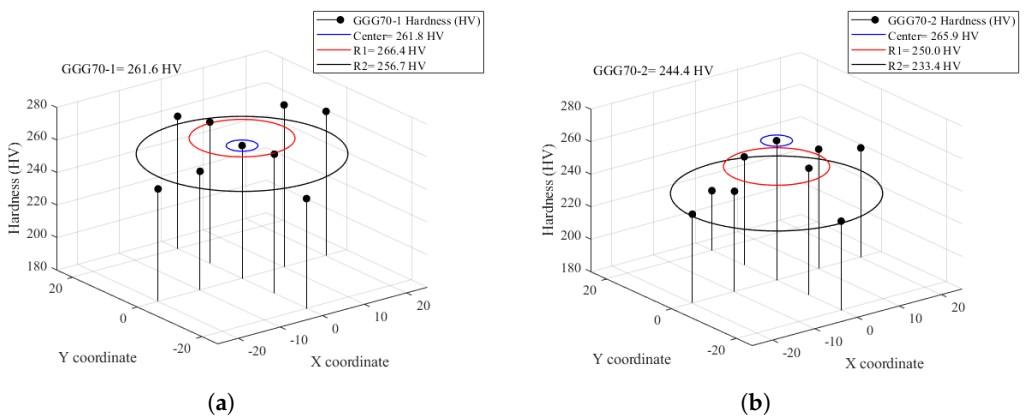

(**a**)                                        (**b**)

**Figure 17.** Hardness test round bar sample. (**a**) GGG70-1. (**b**) GGG70-2.

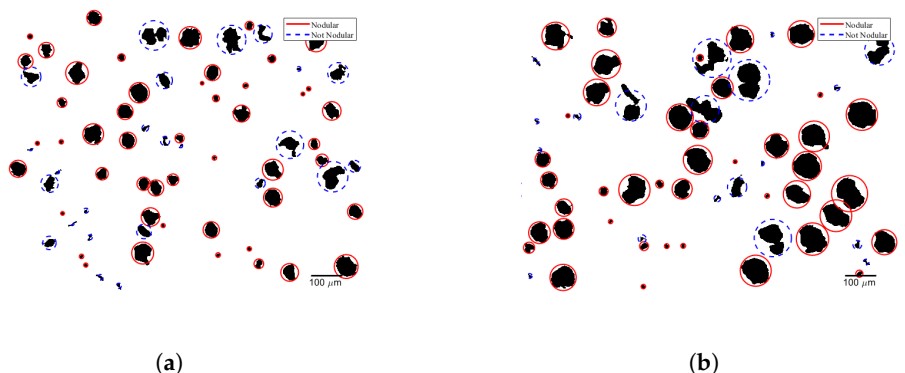

(**a**)                                        (**b**)

**Figure 18.** Computational image identification on GGG40. (**a**) GGG40-1. (**b**) GGG40-2.

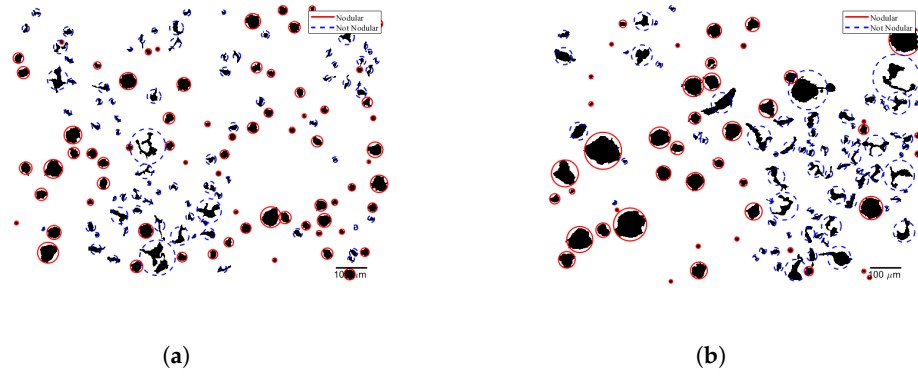

(**a**)                                        (**b**)

**Figure 19.** Computational image identification on GGG60. (**a**) GGG60-1. (**b**) GGG60-2.

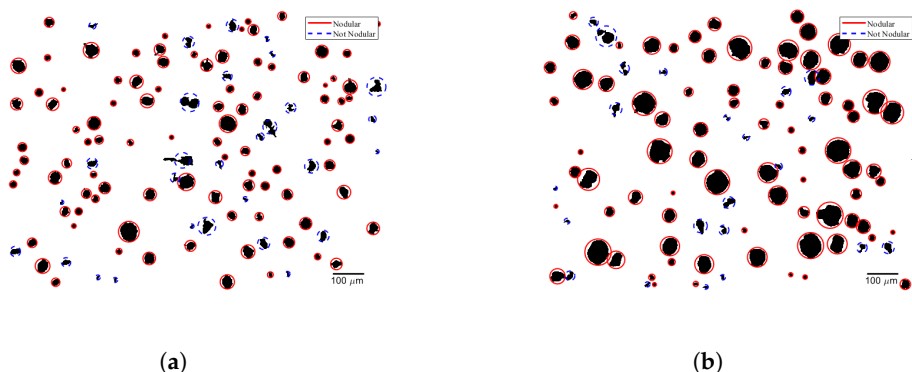

(**a**)                                                                                           (**b**)

**Figure 20.** Computational image identification on GGG70. (**a**) GGG70-1. (**b**) GGG70-2.

From computational image identification figures, the difference between the nodule size in −1 and −2 index samples is noticeable. For all materials, the results were repeated, and the index −2 samples have bigger graphite nodules, which affect the mechanical properties. Another fact that could be analyzed from the image identification is the casting quality of GGG60. In comparison to GGG40 and GGG70, GGG60 has a large number of vermicular particles, which indicates a possible problem in the casting procedure.

In GGG40 and GGG70, the nodules have a satisfactory percentage and shape of graphite nodules. The index −2 samples have better roundness with a slight difference in the shape quality. The number can be verified by Table 4.

Table 4 summarises the graphite characteristics analyzed from round bar specimens.

From Table 4 it is noticeable that the top samples have higher nodule density, although the percentage of nodular graphite in the bottom samples is higher than in the top samples. In accordance with the nodular graphite percentage, the average area of nodular graphite is higher in the bottom samples too. This behaviour is repeated in other NCI samples. The parameters that indicate nodularity, such as sphericity, compactness, roundness, and eccentricity, are slightly close in their pair samples, but when compared overall, the GGG70 specimens present a better cast quality, as can be visually confirmed in Figure 20. The GGG60 samples present a high percentile of graphite flakes (Figure 19), and the data from routine calculations confirm the poor quality through sphericity and eccentricity.

**Table 4.** Round bar specimens graphite characteristics.

|  | GGG40-1 | GGG40-2 | GGG60-1 | GGG60-2 | GGG70-1 | GGG70-2 |
|---|---|---|---|---|---|---|
| Nodule/mm$^2$ | 49.63 ± 12.5% | 36.99 ± 12.9% | 63.40 ± 19.7% | 47.88 ± 29.6% | 107.17 ± 12.4% | 70.59 ± 14.7% |
| Nod. Graphite (%) | 64.71 ± 7.6% | 78.33 ± 8.0% | 58.03 ± 17.0% | 68.21 ± 29.2% | 88.39 ± 5.0% | 90.62 ± 6.1% |
| Avg. Area (μm$^2$) | 1196.4 ± 17.3% | 2576.2 ± 17.9% | 752.1 ±19.8% | 1331.2 ± 19.1% | 751.0 ± 30.3% | 1388.1 ± 12.9% |
| Sphericity | 0.7991 ± 3.6% | 0.8000 ± 3.0% | 0.7534 ± 5.4% | 0.7194 ± 10.1% | 0.9027 ± 1.8% | 0.9057 ± 3.3% |
| Compactness | 0.8678 ± 2.1% | 0.8762± 1.6% | 0.8100± 4.2% | 0.7920 ± 7.8% | 0.9341 ± 1.1% | 0.9356 ± 2.0% |
| Roundness | 0.6543 ± 2.2% | 0.6900 ± 1.9% | 0.5926 ± 6.6% | 0.5832 ± 11.9% | 0.7454 ± 2.4% | 0.7578 ± 2.4% |
| Eccentricity | 0.6720 ± 2.0% | 0.6396 ± 1.7% | 0.7069 ±3.9% | 0.7083 ± 7.2% | 0.5927 ± 3.8% | 0.5776 ±3.5% |

### 3.3. Metallographic and Computational Image Analysis

The developed routine helped us to analyze and quantify the microstructural characteristics of NCI, obtaining values such as nodule density, average graphite size, and roundness. The routine was also capable to quantify the material phases chemically etched with 2% nital, making it possible to obtain values of ferrite, pearlite, and graphite. Figures 21–23 show the computational image analysis of GGG40, 60 and 70 for block specimens and round bar specimens.

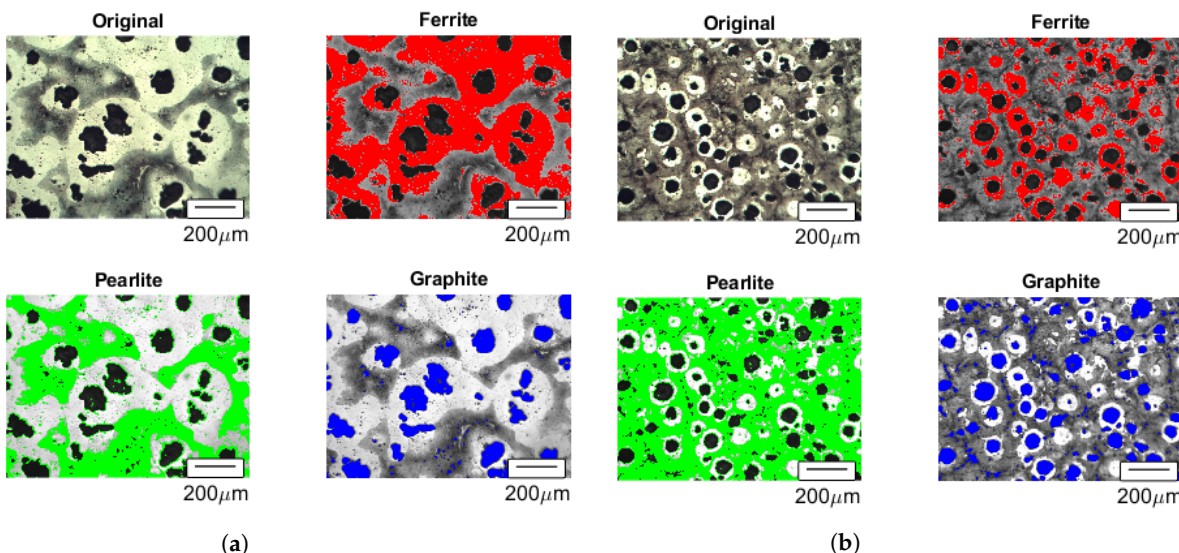

**Figure 21.** Computational image analysis of NCI GGG40 with graphite, ferrite and pearlite identification for round bar specimens. (**a**) Block specimens. (**b**) Round bar specimens.

Comparing Figure 21a,b, the microstructural differences in the samples are remarkable. Even being the same material, casting in different geometries is a significant factor in obtaining the desired microstructure and mechanical properties. The formation of nodules in the cast bar is affected by the cooling rate. The round bar has small graphite nodules but in greater quantity, while the block samples have larger graphite nodules. The amount of the pearlite identified in the circular bar samples was higher than expected. Pearlite values are around 70%.

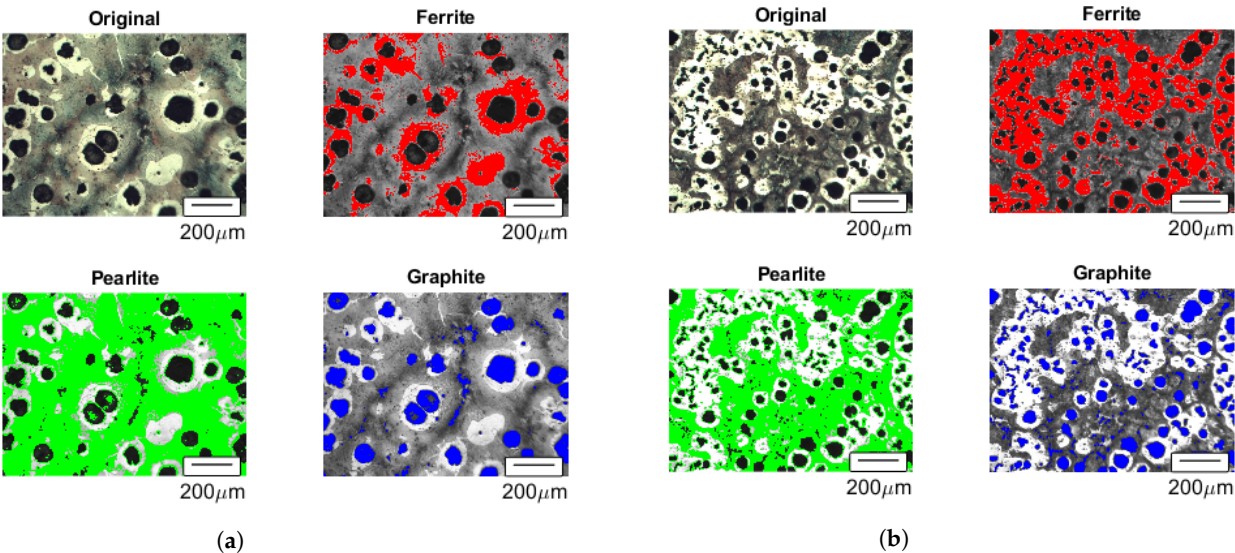

**Figure 22.** Computational image analysis of NCI GGG60 with graphite, ferrite and pearlite identification for round bar specimens. (**a**) Block specimens. (**b**) Round bar specimens.

In the specimens removed from the block, the majority of the graphite particles are considered nodular, which is different from the samples manufactured from round bars that contain a large number of vermicular graphite particles. The round bar microstructure obtained was unforeseen due to the graphite shape and the amount of pearlite in the sample.

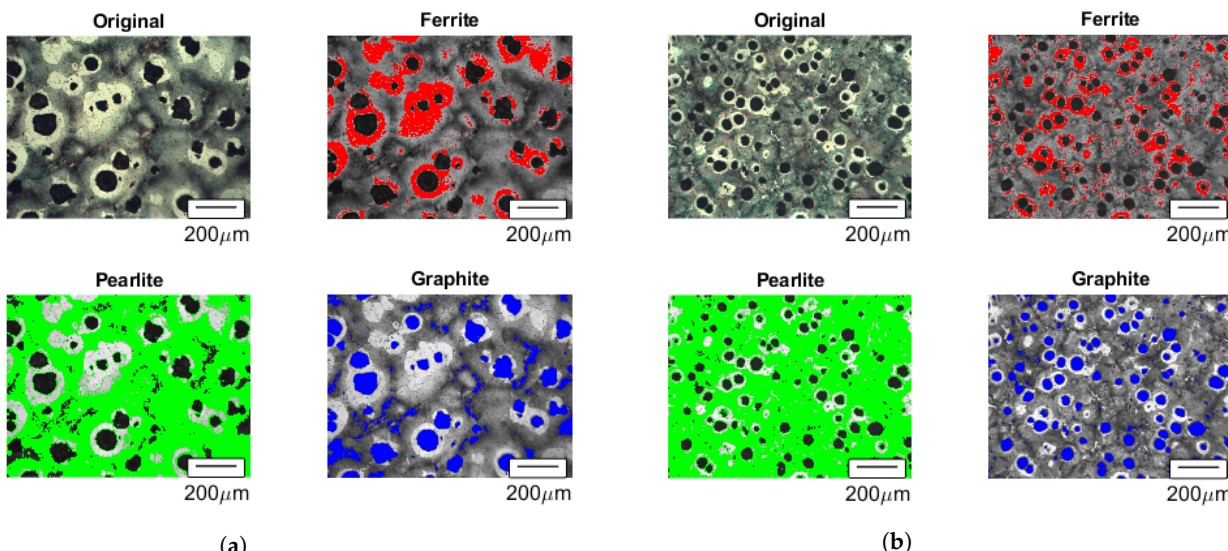

**Figure 23.** Computational image analysis of NCI GGG70 with graphite, ferrite and pearlite identification for round bar specimens. (**a**) Block specimens. (**b**) Round bar specimens.

In GGG70, the majority of the graphite particles have a nodular shape, and the ferrite content is identified around the graphite particles. The green area presents the pearlitic phase and it is clear that the presence of pearlite is greater in the sample, in agreement with the tested material. The ferrite amount in round bar samples is less than in block specimens. This behavior has been seen in other material samples. Round bar samples have more nodular particles with a large number of nodules.

Through the EDS analysis, one could note that Fe was homogeneously distributed between ferrite and pearlite. In addition, a small amount of Fe was noticeable in the center of the graphite nodules. Carbon was seen to be concentrated almost entirely on the graphite nodules. The presence of C was also perceptible on the ferrite and pearlite, but only in small amounts.

Tables 5 and 6 present the results obtained from the developed routine for GGG70 block specimens and round bar specimens, respectively.

**Table 5.** Final results averages for NCI GGG40, 60 and 70 block specimens.

|  | GGG40 | GGG60 | GGG70 |
| --- | --- | --- | --- |
| Hardness (HV) | 197.9 ± 3.0% | 240.7 ± 5.1% | 254.7 ± 4.8% |
| Nodule/mm$^2$ | 46.53 ± 23.5% | 30.89 ± 27.7% | 40.99 ± 42.2% |
| Nod. Graphite (%) | 83.03 ± 2.9% | 93.74 ± 2.7% | 92.09 ± 2.3% |
| Avg. Area (μm$^2$) | 2813.85 ± 21.5% | 4294.90 ± 29.8% | 2870.84 ± 33.6% |
| Sphericity | 0.8825 ± 3.9% | 0.8911 ± 1.3% | 0.8736 ± 3.2% |
| Compactness | 0.9325 ± 1.7% | 0.9343 ± 0.4% | 0.9305 ± 0.6% |
| Roundness | 0.7059 ± 4.2% | 0.7529 ± 1.6% | 0.7491 ± 2.7% |
| Eccentricity | 0.6356 ± 3.0% | 0.5890 ± 2.1% | 0.5875 ± 3.2% |
| Graphite (%) | 13.37 ± 7.0% | 14.26 ± 4.2% | 16.27 ± 11.0% |
| Ferrite (%) | 67.71 ± 7.7% | 27.18 ± 19.8% | 18.50 ± 26.8% |
| Pearlite (%) | 18.91 ± 27.0% | 58.54 ± 9.5% | 65.22 ± 9.8% |

According to [16], roundness, sphericity, and compactness are shape factors that can be used as valid methods for estimating the degree of "roundness" of a graphite particle. These data can provide valid support for a standard classification and routine quality control in ductile iron production. An eccentricity parameter can also aid in the nodule characterization, together with the sphericity shape factor. The eccentricity is a property of the ellipse that best fits the spheroid, as it has values that vary between 0 and 1, describing

how far the graphite element shape is from being circular. Eccentricity and sphericity are the parameters most influenced by varying magnification.

**Table 6.** Final results averages for NCI GGG40, 60 and 70 bar specimens.

|  | GGG40 | GGG60 | GGG70 |
|---|---|---|---|
| Hardness (HV) | 272.4 ± 3.0% | 239.4 ± 10.2% | 275.0 ± 0.7% |
| Nodule/mm$^2$ | 76.88 ± 8.3% | 69.85 ± 12.2% | 71.33 ± 1.4% |
| Nod. Graphite (%) | 90.90 ± 0.7% | 73.30 ± 14.5% | 90.89 ± 1.9% |
| Avg. Area (µm$^2$) | 1551.11 ± 8.6% | 1289.65 ± 20.8% | 1562.91 ± 3.1% |
| Sphericity | 0.9495 ± 1.1% | 0.8371 ± 3.4% | 0.9563 ± 0.9% |
| Compactness | 0.9705 ± 0.7% | 0.8863 ± 2.6% | 0.9749 ± 0.6% |
| Roundness | 0.7738 ± 1.5% | 0.6799 ± 6.2% | 0.7910 ± 0.9% |
| Eccentricity | 0.5678 ± 2.1% | 0.6433 ± 5.5% | 0.5486 ± 1.2% |
| Graphite % | 13.94 ± 9.5% | 13.92 ± 2.7% | 12.95 ± 10.1% |
| Ferrite % | 17.21 ± 21.3% | 24.08 ± 52.1% | 12.44 ± 20.9% |
| Perlite % | 68.85 ± 5.2% | 62.00 ± 20.6% | 74.61 ± 5.1% |

The differences between block-shaped and bar-shaped cast iron are notable, as shown in Tables 5 and 6. The microstructure of bar casting classes has more circular graphite patterns than block casts because the nodules are smaller and do not have enough time to nucleate. GGG60 showed different behavior because it contained a large number of vermicular-shaped graphite particles. A major impact factor was also the percentage of pearlite found in bar-casted samples, being much higher than block samples, especially in GGG40.

A coefficient of variation was used to analyze the dispersion of the results with respect to their mean value. The largest variations found in the measurements resulting from the specimens obtained from the block occurred in the following parameters: nodule density per area, average nodule size, and the pearlite and ferrite phases. For all NCI evaluated herein, those dispersions are justifiable due to the specimen position in the casted block. The specimens located on the block edge tend to have a higher cooling rate, which affects the nodule size and promotes pearlite formation.

In the bar specimens, the largest variation occurs in the average nodule size, and also in graphite, ferrite, and pearlite amount. This phenomenon was noted in both cases, when using block samples and also when using bar samples. It is worth mentioning that the transversal section of the bar sample is thinner when compared to the transversal section of the block sample, resulting in a higher cooling rate for the case of the bar sample. This particular characteristic of the bar sample results in the fact that the shape and sizes of the nodules were more similar, regardless of the sample position on the bar. However, the samples located in the middle of the bar showed higher ferrite levels than the ones obtained at the extreme ends of the bar, as well as slightly larger sizes of the graphite nodules in these central samples.

Table 7 presents the tensile test results for specimens manufactured from the block and the bar.

**Table 7.** Final tensile test results averages for NCI GGG40, 60 and 70 bar and block specimens.

|  |  | $\sigma_{yield}$ (MPa) | $\sigma_{ut}$ (MPa) | E (GPa) | Max. Elong. (%) | $v$ |
|---|---|---|---|---|---|---|
| GGG40 | Block | 413.8 ± 1.8% | 483.3 ± 3.6% | 161.3 ± 5.8% | 3.30 | 0.22 |
|  | Bar | - | 364.5 ± 5.3% | 159.1 ± 10.4% | 0.26 | 0.20 |
| GGG60 | Block | 471.7 ± 1.5% | 506.1 ± 5.2% | 163.5 ± 3.7% | 1.00 | 0.23 |
|  | Bar | 478.5 ± 8.8% | 552.1 ± 2.7% | 146.5 ± 2.1% | 1.84 | 0.21 |
| GGG70 | Block | 488.3 ± 6.5% | 534.5 ± 6.1% | 165.2 ± 5.2% | 1.44 | 0.23 |
|  | Bar | 562.0 ± 2.4% | 650.2 ± 3.8% | 151.0 ± 7.2% | 0.69 | 0.23 |

The differences in mechanical properties obtained in GGG40, 60, and 70 are noticeable, due to the final casting geometry. The specimens of the block presented more uniform mechanical properties when compared to the bar samples. The elasticity modulus values obtained for the block casting material showed good agreement with the literature. On the other hand, the bar samples showed increased strength behavior due to the fact that, in most of the samples, the amount of pearlite obtained was higher in the bar samples.

An important difference between the properties obtained in the tests and the standard properties was noticed in this work. The tests performed reinforce the fact that there is a difference in nodular cast iron properties when obtained from small- and medium-sized foundries, with respect to available standard data. This finding shows the importance of the present study with regard to this class of nodular cast iron materials.

## 4. Conclusions

The present work proposed the analysis and characterization of a batch of commercial nodular cast iron from a Brazilian foundry. It aimed to show that the mechanical properties obtained are not always those available in the literature. The research aimed to evaluate the material as cast in the shape of blocks and bars and to analyze the variety of mechanical properties concerning the obtained microstructure.

It was evidenced by the analysis that material cast in circular bar shapes has different mechanical properties than material cast in a block. The cooling rate is a preponderant factor in obtaining the desirable characteristics. The specimens cast in round bars have a higher pearlite percentage and higher hardness of pearlite and graphite due to the decrease in carbon diffusion. It is important to highlight the increase in the stress due to the nodule density and pearlite amount in the block and bar comparison. According to the literature, the higher the density of the nodules, the greater the strength and elongation capacity of the material. In the bar samples, even with a higher density of nodules, there was only an increase in the strength limit, indicating that this increase was enhanced by the pearlite structure identified in the samples.

The mechanical test performed has evidenced the importance of evaluating the NCI mechanical properties before the material application. It is worth mentioning the strong influence of block Y casting methodology on NCI mechanical properties, especially in the capacity of resisting plastic deformations. NCI has a large variation in mechanical properties from one material to another. In this sense, we evidenced the importance of evaluating the NCI batch before material application. The cast conditions must be very well-controlled in order to obtain a material with the specified properties. The trend graph presents the relationship between the nodule density and the ultimate stress. For all NCI classes analyzed herein, the behavior was similar with respect to the fact that the material tends to resist higher stress levels when the nodule density is greater. Inoculation methods used to increase the number of nodules usually make the nodules more spherical. Therefore, a high number of nodules is usually associated with improved nodularity.

Considering the obtained results, it is valid to continue exploring NCI mechanical properties and how to improve the casting process in order to obtain more accurate properties for the Brazilian foundry. Another suggestion for future work is to determine the mechanical properties of NCI from the microstructure using neural networks and genetic algorithms.

Finally, it can be inferred that the methodology used in this work allows for obtaining NCI 40, 60, and 70 characteristics in the absence of heat treatment and the cast Y block procedure. The procedures aid our understanding of the material's behavior concerning the microstructure. The stipulated objectives were reached, and the developed routine, that seems to be efficient in NCI characterization, will be available on the web.

**Author Contributions:** Conceptualization, D.d.O.F. and C.T.M.A.; Funding acquisition, C.T.M.A.; Investigation, D.d.O.F.; Methodology, D.d.O.F.; Project administration, C.T.M.A.; Resources, C.T.M.A. and J.N.V.G.; Software, D.d.O.F.; Supervision, C.T.M.A. and B.B.; Validation, C.T.M.A., J.N.V.G. and B.B.; Writing—original draft, D.d.O.F.; Writing—review and editing, D.d.O.F., C.T.M.A., J.N.V.G. and B.B. All authors have read and agreed to the published version of the manuscript.

**Funding:** This research was funded by National Council for Scientific and Technological Development (CNPQ) grant (n° 314602/2021-6).

**Institutional Review Board Statement:** Not applicable.

**Informed Consent Statement:** Not applicable.

**Data Availability Statement:** The datasets generated during and/or analysed during the current study are available from the corresponding author on reasonable request.

**Acknowledgments:** This study was financed in part by the Coordination for the Improvement of Higher Education Personnel—Brazil (CAPES), Brazilian Council for Scientific and Technological Development (CNPq) and Research Support Foundation of Federal District (FAPDF). The authors are grateful to Group of Experimental and Computational Mechanics (UnB/FGA/GMEC) for providing computational resources and making this work possible. The second author thanks the National Council for Scientific and Technological Development (CNPq).

**Conflicts of Interest:** The authors declare no conflict of interest.

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
