# Peer review of "Nodular Cast Iron GGG40, 60, 70 Mechanical Characterization from Bars and Blocks Obtained from Brazilian Foundry"

_metals, doi:10.3390/met12071115_

Round 1

Reviewer 1 Report

The paper regards the correlation between the mechanical properties of different grades of nodular cast irons and the characteristics of the graphite nodules.

Bibliography is not present, so it was not possible to check the cited papers.

According to my point of view several general aspects should be improved in the paper before to be considered for publication in Metals. Accordingly, a detailed review of the work should be also performed.

  • In Section 1 “Introduction” the objectives of the work must be explained better. The aim of the work should not be mentioned in Section 2 “Materials and Methods” (line 84-85). Moreover, in the introduction the novelty of the work with respect to the bibliography should be explained.
  • It is not clear while the authors did not consider the “metallographic procedure” in Section 2 “Materials and Methods”. Metallographic procedure is part of the methodology. Please improve this point.
  • In Section 2 “Materials and Methods” the authors showed in Table 1 the chemical compositions of the alloys. Please specify how the compositions were determined. Moreover, the paragraph “2.1 Materials” is quite confusing and not clear for the reader: please, improve.
  • In Section 2 “Materials and Methods” the authors did not say they also performed cyclic tests, but in the following of the paper they present the results of this kind of tests. Please, improve.
  • Concerning the paragraph “2.3 Hardness test”, the load applied for hardness and microhardness tests should be clearly specified. Currently, they reported just a range. Talking about the results of hardness tests (see Table 3), they just reported mean values, but not standard deviations or standard errors. How many tests were performed for each conditions and in each zone?
  • As concern the analysis of graphite, with reference to Figure 2, how many micrographs for each zone were acquired and analysed?
  • Please add scale bar in Figure 3.
  • Figure 4 must be described better. Please add some comments about the flowchart procedure.
  • The authors decided to create different sections for “Results” and “Discussion”, but several times in the Section 4 “Results” the mixed results and discussion; for example, see Figure 7 where they correlate nodule densities and maximum stresses. Moreover, before Figure 7 they did not present the results of nodule densities. It is worth noting that Section 5 “Discussion” is quite poor, so it must be improved and a clear separation between results and discussion must be carried out.
  • In Section 4 “Results” a table summarizing the results of tensile tests should be added, also with standard deviations or standard errors.
  • In Section 4 “Results” please add scale bars in all the micrographs of Figures 21, 22 and 23.
  • In addition to the comments reported in 9), in Section 5 “Discussion” the authors should add graphs and/or diagrams which should really help the reader to understand the correlations they found among all the mechanical properties and the microstructural features (nodule density, morphology, ecc).
  • The Section “Conclusion” should be revised accordingly

Author Response

Dear reviewer, 

The authors would like to thank in advance for all contributions.

My best regards.

Reviewer 2 Report

The mechanical properties of nodular cast iron was determined by its microstructure and the shape of graphite. This work evaluated and characterized the nodular cast iron grades GGG40, GGG60 and GGG70 in the absence of heat treatment.the microstrucre is characterized by optical microscopy with the support of computational image. The results is predirectable. Eventhough, the works is systematic. But the creative of this work is not very impressive. There are some suggestions for the author for improve its work.

  • Please tell us the cerative of this work.
  • Since the mcirostrcuctre of the nodular cast iron is not uniform, pease tell us the error  bar of the hardenss, if your could use miro-hardeness tester or nanointedntion to obtain the hardenss of different phase maybe better.
  • The mechanism of the mcirostructure of nodular cast iron with its microstrcutre have been well konwn ,what is new here?
  • In Computational image analysis,please specify the details.
  • The difference of properties of  Block and Round Bar samples may be also from the difierent matrix, ferrtie and perlite. How to sperate the effect of graphites density and matrix on its properties?

Author Response

(The authors gave the same response as above.)

Reviewer 3 Report

Dear Authors,

You made a huge effort to conduct your experiments. However, I must say, your results do not add anything new to knowledge about the nodular cast iron. Also, I did not understand what the main idea was and the aim of the experiments. 

I found a big problem with the references and citations that I reported to the MDPI office, but I have not received an answer so far. Instead of the reference number, there are [?] characters and no reference list at the end of the article. So I was unable to check this aspect of the work. Below I put my comments and unfortunately my final decision is negative. I do not recommend the article for publication. If you think of the idea again and rewrite the article, I encourage you to submit it again as the new submission. 

1 The abstract suggests that there is nothing novel in the article. Everything mentioned there has been obvious for many years.

  1. Line 19: the abbreviation NCI is not commonly used in the literature, however, it is not a mistake.
  2. Line 43: the brake disc is not a typical application of the nodular iron.
  3. The Introduction mostly gives very well-known data. Reading this, I did not feel like I would find anything novel in the article.
  4. Line 86: I don’t like the style of this sentence.
  5. Line 102: I do not understand the idea of casting such blocks. Why? There are standards for casting the samples, and they are required by the clients. What was the shape of the mold? The gating system? The pouring method? These are very important issues for casting, particularly for such bulky ones. What about the riser, etc. It should be explained.
  6. Line 111 claims that the test was based on the ASTM standard but, as I mentioned above, the sampling method is different from that commonly used.
  7. Line 110: Traction test: What does it mean?
  8. Figure 2: What was the place to cut the metallographic samples? I guess they were cut from the bars shown in Figure 1? But from which part exactly? The microstructure of such samples is not representative of the entire block, especially the mold itself.
  9. Figure 3: There is neither the magnification nor the scale bar presented there.
  10. Figure 4: I do not think it is necessary to explain the software algorithm here. The image analysis software for such purposes is well known.
  11. Line 168: The shape factor 0.6 is quite low to claim that the graphite precipitate is a nodular one. Typically, it is around 0.8 or higher.
  12. Line 194: I do not understand how the authors may claim that the lack of quality control was the reason? It was a scientific experiment.
  13. Table 3: The microhardness values seem to be too high in my opinion. Check it out.
  14. Line 264: The solidification process and the final microstructure depend greatly on the geometry of the casting, the entire gating system, and the pouring process. Without information about them, it is difficult to discuss the issue. I mentioned that earlier.
  15. Figs. 18-20: The nodule size, shape, and distribution are far from perfect and I suppose they will fail the quality control in most foundries.
  16. Conclusions: reading them, I was still not sure what the idea of these experiments was?

Sincerely,

Reviewer

Author Response

(The authors gave the same response as above.)

Round 2

Reviewer 1 Report

The authors performed most of the suggested revisions, but I have some more considerations.

  • Concerning Table 1 “Chemical composition of the alloys”, the authors say that they used an SEM/EDS technique which is a semi-quantitative analysis and light elements, such as Carbon, are difficult to be determined. Did they use a standard before performing the analysis? What is the scatter of data? Please specify.
  • The quality of Figure 2b should be improved.
  • The author should discuss better the correlation between the microstructure parameters (nodule density, morphology, etc) and the mechanical tests. For example, Tables 4, 5 and 6 present a lot of geometrical parameters (roundness, eccentricity, compactness, etc), but the authors do not deeply discuss these results with respect to mechanical properties. If all these geometrical parameters are important/relevant, the author should try to discuss if there is a significant correlation between these parameters and the mechanical properties (σy, σut, A%). The values reported in those tables refer to mean values; what about the distribution of the values? Do they follow a normal distribution or not? In the first revision of the paper, my comment on the “discussion” section was also about this kind of improvement. So, I hope the authors can clarify these points and improve the discussion section accordingly.
  • According to my point of view, English should be checked by a mother tongue before submitting the revised version of the paper.

Author Response

Dear Reviewer

My best regards.

Reviewer 2 Report

Authors made sufficient work to meet the suggestions of the reviewers.

Author Response

Dear Reviewer,

My best regards.

Reviewer 3 Report

Dear authors,

I see that you have made an effort to improve your article. The result is positive and now the article looks much better. Therefore, I recommend it for publishing as is.

Sincerely,

Reviewer

Author Response

Dear Reviewer,

My best regards.

Round 3

Reviewer 1 Report

Dear authors,

you have made of your best to improve the paper according to the suggestion of the reviewers.